# Immune Tumor Microenvironment in Ovarian Cancer Ascites

**DOI:** 10.3390/ijms231810692

**Published:** 2022-09-14

**Authors:** Diana Luísa Almeida-Nunes, Ana Mendes-Frias, Ricardo Silvestre, Ricardo Jorge Dinis-Oliveira, Sara Ricardo

**Affiliations:** 1Differentiation and Cancer Group, Institute for Research and Innovation in Health (i3S), Institute of Molecular Pathology and Immunology, University of Porto (IPATIMUP), 4200-135 Porto, Portugal; 2TOXRUN—Toxicology Research Unit, University Institute of Health Sciences, CESPU, CRL, 4585-116 Gandra, Portugal; 3Life and Health Sciences Research Institute (ICVS), School of Medicine, University of Minho, 4710-057 Braga, Portugal; 4ICVS/3B’s—PT Government Associate Laboratory, 4710-057 Braga, Portugal; 5UCIBIO-REQUIMTE, Laboratory of Toxicology, Department of Biological Sciences, Faculty of Pharmacy, University of Porto, 4099-002 Porto, Portugal; 6Department of Public Health and Forensic Sciences, and Medical Education, Faculty of Medicine, University of Porto, 4099-002 Porto, Portugal; 7MTG Research and Development Lab, 4200-604 Porto, Portugal; 8Faculty of Medicine, University of Porto (FMUP), 4099-002 Porto, Portugal

**Keywords:** ovarian cancer, malignant ascites, tumor microenvironment, immune cells, cytokines, high-grade serous carcinoma

## Abstract

Ovarian cancer (OC) has a specific type of metastasis, via transcoelomic, and most of the patients are diagnosed at advanced stages with multiple tumors spread within the peritoneal cavity. The role of Malignant Ascites (MA) is to serve as a transporter of tumor cells from the primary location to the peritoneal wall or to the surface of the peritoneal organs. MA comprise cellular components with tumor and non-tumor cells and acellular components, creating a unique microenvironment capable of modifying the tumor behavior. These microenvironment factors influence tumor cell proliferation, progression, chemoresistance, and immune evasion, suggesting that MA play an active role in OC progression. Tumor cells induce a complex immune suppression that neutralizes antitumor immunity, leading to disease progression and treatment failure, provoking a tumor-promoting environment. In this review, we will focus on the High-Grade Serous Carcinoma (HGSC) microenvironment with special attention to the tumor microenvironment immunology.

## 1. Introduction

Ovarian cancer (OC) is one of the most frequent gynecologic cancers, being the seventh most-common malignant tumor and the eighth cause of cancer death in women around the world [1]. This malignancy is characterized by the rapid growth and spread of multiple intraperitoneal tumors [2]. The World Health Organization classifies ovarian tumors as epithelial, which represents almost 90% of the cases, germ cell (3%), and sex cord-stromal (2%) origins [3]. Epithelial Ovarian Carcinoma (EOC) has five main subtypes based on its histopathology, immune, and molecular profile: high-grade serous carcinoma (HGSC) comprising 70% of total EOC, low-grade serous carcinoma (LGSC), endometrioid carcinoma (EC), clear cell carcinoma (CCC), and mucinous carcinoma (MC) (Figure 1) [3]. Despite the differences between the subtypes, these tumors are treated as a single entity, relying on cytoreductive surgery and platinum-taxane combination chemotherapy. The response rate to first-line therapy is about 80–90%, but most patients relapse and develop chemotherapy resistance, leading to a 5-year survival rate below 25% [4,5]. Serous carcinomas are tumors with a more aggressive behavior, e.g., HGSC spreads rapidly throughout the pelvis, while endometrioid and mucinous carcinomas are typically confined to the ovary being mostly low-grade lesions [6]. The histologic types of OC are highly associated with the clinical response. Late-stage serous and clear cell carcinomas, both have similar 5-year survival rates (20–30%); however, their response to first-line chemotherapy is different: Serous tumors are highly responsive, while clear cell tumors are especially resistant [7]. However, the survival rate of patients with serous tumors relies on chemoresistance upon recurrence which is very frequent.

OC has a lower prevalence in contrast to breast cancer but is three times more lethal due to the lack of specific symptoms and the absence of good screening tools, resulting in diagnosis at advanced stages [8]. OC patients present high recurrence rates, mostly due to the incomplete resection of tumors in cytoreductive surgery [9]. In addition, although most patients are platinum-responsive to first-line therapy, most of them develop chemoresistant recurrences [9]. These facts contribute to worse clinical outcomes; approximately 15% of OC patients die from the disease within the first year, and only 25% survive over 5 years from the date of diagnosis [10]. These different outcomes trigger the need for a more profound exploration of tumor and host characteristics. The anti-tumoral immune response in OC patients could help to disclose these different outcomes. OC is an immune reactive malignancy in which tumor cells establish a complex multilayered immune suppression network that effectively neutralizes most attempts to increase antitumor immunity [10]. This leads to a lack of responses to immune-based treatments in these patients.

In this review, we present a summary of the immunological populations and their role in OC, including the subsets of adaptive and innate immune response, the mechanisms of immune escape, and the effect of key cytokines, chemokines, and growth factors in the tumor immune microenvironment (TIME).

## 2. High-Grade Serous Carcinoma

HGSC is the most common ovarian malignancy, representing approximately 70% of ovarian carcinoma [3]. It is recognized to be of epithelial origin, arising from the fallopian tube epithelium through a series of precursor lesions that target the secretory cell [7,11]. This tumor type is generally recognized by the lack of architecture and sheets of malignant cells with enlarged and dysmorphic nuclei, and the high frequency of the tumor suppressor p53 protein (TP53) mutation [12]. The TP53 mutation occurs exclusively in secretory cells that show strong expression of this gene and evidence of DNA damage but are not proliferative. The progression to a serous tubal intraepithelial carcinoma involves the gain of nuclear pleomorphism and mitoses and the loss of cell polarity [7,8].

The median age of patients diagnosed with this type of tumor is 56 years (ranging from 45 to 65 years) [3]. The available screening techniques are unsuccessful for early detection and just a minority of OC patients are diagnosed with tumors limited to one ovary (<5%). Typically, HGSC presents at diagnosis with bilateral ovarian involvement and diffuse and extensive peritoneal carcinomatosis, particularly with omental involvement, characterized by the presence of malignant ascites (MA) [13,14]. The spread of multiple intra-abdominal tumors is often associated with signs of intestinal obstruction, including nausea, vomiting, persistent bloating, and abdominal pain. Ultrasound, magnetic resonance imaging, and computerized tomography have no specified role in preoperative tumor staging. Thus, laparotomy and the surgical exploration of the abdominal cavity are used as standard approaches to defining the tumor stage [3]. Although specific biomarkers for OC are not defined, women with deleterious germline *BRCA1* or *BRCA2* mutations have a 30–70% risk of developing HGSC [3,15].

Despite these features at diagnosis, HGSC patients are usually highly sensitive to first-line therapy (platinum-taxane) and only a small subgroup (<10%) is refractory to this chemotherapy [16]. Nevertheless, even after a period of initial clinical remission, most patients suffer from a relapse of the disease (approximately 75%) in 3 years that are typically incurable [16,17]. Some of these patients are refractory to first-line therapy due to acquired chemoresistance, while the majority suffer remission with the same treatment. This fact shows a potential mechanism of therapy failure different from intrinsic or acquired resistance [18].

## 3. Malignant Ascites—Tumor Microenvironment in Ovarian Cancer

Ascites can occur in different diseases, such as cirrhosis, pancreatitis, nephritis, heart failure, and cancer [19]. MA are the pathological accumulation of fluid in the peritoneal cavity that can be found in many neoplasms of peritoneal organs (ovarian, endometrial, pancreatic, gastric, colorectal, and liver) [20]. This inflammatory condition is a consequence of a disruption in the balance of fluid production and reabsorption [21], being triggered by increased vascular permeability, peritoneal lymphatic obstruction, and high levels of fluid production [22]. The presence of MA is often indicative of tumor cells in the peritoneal cavity or peritoneal carcinomatosis [23].

Several studies associate OC growth and metastization with its intrinsic tumor microenvironment [24,25,26]. The late-stage diagnosis (stage III/IV) implies the presence of metastasis in the pelvic and peritoneal cavities which are coupled with the accumulation of a large volume of peritoneal fluid [27]. Unveiling the mechanisms behind this liquid metastatic microenvironment is crucial for the improvement of new attempts to abrogate the tumor peritoneal spread and improve OC management.

The role of MAs is to facilitate the spread of tumor cells to other pelvic and peritoneal organs, serving as a transporter [28]. This transcoelomic dissemination process is fundamental for the adhesion of cancer cells to the omentum and serous membranes lining the peritoneal organs, leading to the implantation of metastatic tumors in the peritoneal cavity, not being typical the invasion of the lamina propria [29]. OC cells disseminate into peritoneal sites such as the hepatic hile, omentum, spleen, and uterus, among others. MA comprise not only tumor cells, but also many other non-tumor cells (Figure 2), which create a unique microenvironment capable of modifying the neoplastic properties of tumor cells [30]. This exudative fluid is composed of a cellular and an acellular fraction. The cellular counterpart is composed of highly tumorigenic cancer cells [31,32], innate and adaptive immune cells [10,22], and other non-tumor cells. The acellular fraction contains tumor-promoting soluble factors (angiogenic and growth factors), bioactive lipids, cytokines, and extracellular vesicles [33]. All of these factors contribute to tumor cell proliferation, progression, chemoresistance, and immune evasion [34], suggesting that MA play an active role in the development and progression of OC, especially in HGSC [35]. Stromal cells can regulate the extracellular matrix (ECM) composition and produce molecules that could attract ovarian carcinoma cells to bind to the ECM [36,37]. HGSC is typically highly vascularized, which correlates with a poor prognosis [30,38]. Ascites-associated cancer cells appear as single cells or multicellular spheroids and are responsible for peritoneal dissemination and disease relapse [39]. The multicellular spheroids are crucial mediators of metastasis, and their survival in the MA environment is essential for peritoneal dissemination because this type of cellular organization allows OC cells to resist anoikis and apoptosis induced by chemotherapeutic agents [27].

### 3.1. The Tumor Immune Microenvironment

OC cells and the TIME components maintain an important crosstalk, which is involved in reprogramming both innate and adaptive immune responses and promoting tumor growth and metastasis. The tumor microenvironment communication consists of an indirect cytokine-mediated interaction and a direct cell–cell interface between cancer cells and stromal cells. As a consequence of this communication, several pathways function as potent regulators contributing to the aggressive and metastatic footprints of HGSC [16]. MA contain an environment of pro-inflammatory factors that contribute to the release of mucin 16 (MUC16), a well-known glycoprotein involved in OC tumorigenesis and metastasis [40,41,42,43,44,45,46,47,48]. Pro-inflammatory cytokines and chemokines are the key element of MA and modulate HGSC in paracrine and autocrine manners [49]. This inflammatory response towards cancer cells supports the infiltration of neutrophils, which promotes cancer progression via the secretion of transforming growth factor-β (TGF-β), tumor necrosis factor (TNF)-α, metalloproteinase 9 (MMP9), reactive oxygen species (ROS), and nitric oxide (NO) [50]. The production of ROS and NO inhibits T cell infiltration and activation, which leads to T cell apoptosis [50]. In connection with this, a higher neutrophil-to-lymphocyte ratio (NLR) is correlated with decreased overall survival in OC patients [51,52].

In the OC context, MA promote an immunosuppressive microenvironment with cellular immune populations that do not reflect the populations present in the patient’s blood or tumor [53]. The immune cells are classified into two categories: anti-tumoral (cause tumor cell death) and pro-tumoral (tumor-promoting cells) (Figure 3). The anti-tumoral cells include cytotoxic T lymphocytes (CD8^+^) and activated helper T cells (CD4^+^). The pro-tumoral cells are myeloid-derived suppressor cells (MDSCs), tumor-associated macrophages (TAMs, especially pro-tumoral phenotype), lymphocyte T helper cells (Th2 subtype), and T regulatory cells (Tregs) [16]. In fact, the presence of myeloid cells, such as MDSCs and immature dendritic cells (DCs), is usually found in tumors [54,55]. There are several studies that strongly support the theory that promoting anti-tumoral responses could be crucial for controlling the tumor progression of HGSC [56].

#### 3.1.1. Innate Immune Cells in the Ovarian Cancer Tumor Immune Microenvironment

Myeloid-Derived Suppressor Cells (MDSCs)

Myeloid cells are frequently observed in the stroma of tumors [57] and, in the OC context, they present an increased capacity to block local and systemic immune activation [58]. MDSCs are composed of a heterogeneous population of immature myeloid cells (macrophages, DCs, and granulocytes at initial stages of differentiation [59]) that grow in pathologic conditions and have a high potential to suppress T cells [60,61,62]. These cells are important inducers of tumor immune evasion and impaired immunity by upregulating arginase-1 and generating NO and ROS [58]. MDSCs also deplete cysteine, induce Tregs, inhibit T-cell activation and proliferation, reduce the cytolytic ability of NK cells, and trigger a pro-tumoral phenotype in macrophages [63]. In ten pre-clinical models of tumorigenesis, MDSC subgroups were associated with immune suppression [64]. The contribution of MDSCs who express the combination markers (Lin-CD45^+^CD33^+^) was studied in a cohort of patients with HGSC [65], and the results showed that 37% of non-neoplastic cells were MDSCs in the TIME, being also responsible for inhibiting T cell immunity, by blocking both T cell proliferation and effector function. In this study, the authors also showed that an increased tumor of MDSCs is negatively correlated with CD8^+^ tumor-infiltrate lymphocytes (TILS) and overall survival in advanced OC [66]. Curiously, the same populations of MDSCs in patients’ blood have distinct functions that they have in MA, suggesting that they support metastasis and a cancer stem cell phenotype in MA. Mechanistically, it is proved that tumor-resident MDSCs increase cancer cell stemness through the upregulation of microRNA101, which targets the co-repressor gene C-terminal binding protein-2 (CtBP2) 3′-UTR region and interferes with its binding at NANOG, OCT4/3, and SOX2 promoters in primary OC cells [65]. Additionally, the concentration of MDSCs is correlated with poor patient prognosis and elevated levels of IL-6 and IL-10 [66,67], in addition to VEGF expression and the production of adenosine by OC cells that induce MDSCs recruitment, inhibiting local immunity [66,68]. This evidence supports immunotherapeutic strategies targeting MDSCs that could help to improve antitumoral responses. For example, it is possible to block MDSC suppressor functions by decreasing the expression of CD39 and CD73 using metformin, a drug for type 2 diabetes. This blockade will promote HGSC clinical benefits by improving antitumoral T cell responses that were inhibited by MDSCs in the TIME [69]. This shows a strategy to reduce tumor progression by targeting immature myeloid cells and their crosstalk with other immune cells and cancer cells. Several drugs targeting MDSCs or tumor-associated macrophages (TAMs) have been described [69] and they include inhibitors of immune suppression function (sildenafil, triterpenoids, COX-2 inhibitors, nitric oxide inducers), antibodies that stimulate the depletion of MDSCs and/or TAMs, blockers of recruitment (by targeting chemokines and their receptors) or MDSCs proliferation, promoters of MDSCs apoptosis, TAM reprogramming factors, and inducers of immature myeloid cells differentiation (such as retinoic acid or vitamin D3) [70]. TAMs and MDSCs promote resistance to targeted immunotherapy, so repressing these populations could increase the success rate of checkpoint blockade inhibitors such as nivolumab and pembrolizumab [71].

Macrophages

Monocytes develop from myeloid cells found in the bone marrow and after maturation, they circulate in the bloodstream and can migrate into tissues where they differentiate into macrophages [72]. The macrophages can be polarized in separate ways, express specific surface markers, and acquire distinct functional states, depending on stimulating factors such as cytokines and other signals [29]. Activated macrophages are divided into “M1” and “M2” types based on in vitro phenotypes. Classically activated macrophages, also known as M1-like, are polarized by pro-inflammatory stimuli such as interferon (IFN)-γ and lipopolysaccharides (LPS), and they release TNFα, IL-1β, IL-6, IL-12, and IL-23, exhibiting anti-bactericidal, immunostimulatory, and antitumoral activities [29,72]. Alternatively activated macrophages, or M2-like, are polarized by anti-inflammatory stimuli such as IL-4, IL-10, and IL-13 and they release IL-6, IL-10, IL-13, TGF-β, VEGF-A, arginase, normally related with wound healing, cellular proliferation, the resolution of inflammation, immunosuppression, tumor invasion, tumor growth, angiogenesis, and metastasis [29,72]. Hence, M1-like macrophages inhibit tumor growth and M2-like are pro-tumoral [72]. Hagemann et al. described that OC cells have the capability to regulate the macrophage phenotype. In fact, OC cells can differentiate macrophages into a TAM phenotype [73]. The work of Duluc et al. demonstrated that this process is caused by the actions of the leukemia inhibitory factor (LIF), IL-6, and the macrophage colony-stimulating factor (M-CSF) [74]. TAMs present in the omentum predominantly have an anti-inflammatory phenotype to facilitate tumor progression [75] by secreting cytokines such as IL-6 and IL-8 [76]. Yin et al., [77] showed that TAMs were localized within spheroid centers and secreted EGF, by the signaling pathway EGF–EGFR [78], causing an upregulation of integrins and VEGF signaling by the activation of the NF-κB and JNK signaling pathways [79], supporting both tumor cell proliferation and migration. Macrophages have a dual effect on tumor cells: on the one hand, they can increase cancer cells’ invasion potential by TNFα and NF-κB pathways; on the other hand, they can facilitate the OC dissemination of tumors in the peritoneum by combination with VEGF, proteases, and secreted growth factors [80].

Chemotherapy also modulates macrophage activity, promoting an inflammatory environment, which paradoxically promotes tumor growth, mediated by the release of bioactive lipids from macrophages, which stimulate cyclooxygenase (COX) pathways [81]. Reader et al. [82] showed that chemotherapy-resistant cancer cells that overexpress class III β-tubulin in response to taxanes are originated by the inhibition of EP4 receptors, leading to a downstream product of COX enzymes, Prostaglandin E2 (PGE2). EP4 is overexpressed in various EOC histotypes [83]. The inflammatory stimuli of the environment in MA activate the upregulation of class III β-tubulin [84], being associated with a more aggressive biologic OC behavior [23,84]. Macrophages in MA are distinguished by the expression of surface markers: CD163^+^ is associated with a recurrence-free survival [22]; CD163 is a scavenger receptor that affects hemoglobin–haptoglobin complexes but also interacts with erythroblasts and may be distorted to a pro-tumoral phenotype [85]. The macrophage-derived chemokine (MDC=CCL22) is secreted by the macrophages and OC cells of MA, which attract Tregs to the tumor, suppressing T cell immunity and enhancing tumor growth [86]. The B7-H4 is a costimulatory molecule that decreases the proliferation and cytokine production of T cells and is expressed by a subpopulation of OC stromal macrophages, so their presence correlates with the number of tumor-infiltrating Tregs, which negatively regulate T cell immunity, leading to a poor outcome in OC [87]. Recapitulated, TAMs could be considered markers of poor prognosis since there is a clear association between the abundance of TAMs and tumor progression [74].

Neutrophils

Neutrophils are leukocytes specialized in phagocytosis and defense against invading microorganisms. Although their antibacterial functions are well described, there is a rising interest in their role in the cancer context. In fact, intratumoral neutrophils can be separated into anti-tumor, “N1-like”, and pro-tumor, “N2-like” [88]. Based on the tumoral context, tumor-associated neutrophils (TANs) can present a different phenotype and show either N1-like or N2-like phenotypes [89]. Although there are no studies showing a specific function of TANs in OC progression, Lee et al. demonstrated that OC cells release IL-8, leading to decreased tumor growth which could be in part attributed to the recruitment of neutrophils [90]. In another study, Klink et al. demonstrated that the direct interactions between neutrophils and OC cells stimulated increased ROS production, adhesion ability, and the upregulation of CD11b/CD18 expression in neutrophils from OC patients compared with neutrophils from healthy woman volunteers [91]. The NLR has been pointed to as an indirect measurement of inflammatory status and several studies showed that elevated NLR is a prognostic factor associated with an increase in disease recurrence in several cancers [92,93,94,95,96]. A study by Cho et al. evaluated the prognostic significance of NLR in patients with OC compared with patients with benign gynecological tumors and healthy controls [93], and showed that OC patients with high preoperative NLR had a decreased overall survival compared to patients with low NLR [93], meaning that neutrophils have a potential immune deregulating role in OC.

Dendritic cells

DCs are professional antigen-presenting cells (APCs) that link innate and adaptive immunity and are critical for the induction of protective immune responses [30]. DCs are classified into two subtypes according to their lineage: plasmacytoid DCs (pDCs) from the lymphoid lineage, expressing CD123, CD45RA, CD8, and ILT3, and myeloid DCs (mDCs) from the myeloid lineage, expressing CD11c and CD33 [54]. Under inactive conditions, these cells wander in the body in an immature form and are responsible for detecting phagocyte pathogens. DCs are activated by their pathogen-associated molecular pattern (PAMP) receptors or danger-associated molecular pattern (DAMP) receptors, and then they mature and migrate to the lymph nodes to activate CD4^+^ and CD8^+^ T lymphocytes [54]. Although tumors produce danger signals, they are useless in inducing DC maturation, leading to a reduced number of functional cells available for effective T-cell activation [97]. Wei et al. showed that tumor-associated pDCs (TApDCs) can modify ovarian tumor immunity by inducing immunosuppressive CD8^+^ T lymphocytes [98]. Similarly, Curiel et al. showed that ovarian tumors can reject mDCs, which have angiogenesis inhibition properties, and attract pDCs, which enhance angiogenesis via TNFα and IL-8 secretion [99]. In OC, some TApDCs (CD11^+^) acquire endothelial and pericyte characteristics and participate in the preservation of tumor vasculature. In fact, the depletion of these cells results in vascular apoptosis, tumor necrosis, and an increased result of chemotherapies and anti-tumor immunity [100]. Labidi-Galy et al. found phenotypic and functional differences between TApDCs and pDCs in advanced OC, supporting the theory that pDCs exhibit pro-inflammatory properties, whereas TApDCs have strong immunosuppressive characteristics and correlate with early relapse and a poor outcome [101,102]. MAs are enriched with pDCs but not mDCs [103], which, curiously, is not correlated with the survival of HGSC patients [103], suggesting deficits in functionality. This lack of functionality could be correlated with the expression of PGE2 and its receptors (EP2 and EP4) by DCS, which comprises Toll-like receptor-mediated DC activation [104], suggesting one mechanism by which the inflammatory environment of MAs may lead to DC malfunction.

NK cells

NK cells are key effectors in cancer immunosurveillance, recognizing and spontaneously killing virus-infected cells, cancer cells, and pathogens [105]. NK cells secrete pro-inflammatory cytokines and chemokines such as IFNγ, TNFα, IL-6, granulocyte-macrophage colony-stimulating factor (GM-CSF), and C-C Motif Chemokine Ligand (CCL5), promoting antitumoral innate and adaptive responses in the TIME [106]. Ovarian carcinoma effusions were analyzed and show the presence of NK cells in advanced stages (IV), predicting a worse overall survival [107]. Nevertheless, it has been reported that NK cells along with effector CD8^+^ T cells have a positive antitumoral role [108], and the activity of circulating NK cells was related to a significant progression-free survival of OC patients [109]. Like many other cancers, ovarian carcinoma tumors express the NK cell receptor ligand ULBP2, which is an indicator of poor prognosis and could promote T-cell dysfunction in the TIME [110]. Vazquez et al. [111] found that an increase in cytolyzing cancer targets of CD56^bright^ NK cells is not correlated with an increase in producing cytokines. Contrarywise to this, CD16^+^ NK cells are associated with cytotoxic responses but are significantly reduced in HGSC MA [111]. A high density of NK cells is found in MAs but often without functionality [112]: in these cases, IL-18 and TGF-β decrease CD16 expression in NKs, impeding antibody-dependent cellular cytotoxicity [113,114]. A reduced number of NK cells in MAs also correlates with chemoresistance [53]. Essentially, NK cells are activated or not, according to the equilibrium between inhibitory and activating signals through different NK receptors [114]. NK cells also express estrogen receptors and programmed death protein (PD)-1 [115], suggesting additional mechanisms by which hormonal modulation and checkpoint inhibition may affect OC. Nham et al. realize a study with an artificial APC-based ex vivo expansion technique to generate cytotoxic expanded NK cells for use in an autologous model of immunotherapy [116]. In this study, they acquire NK cells from the MAs of OC patients that upregulated the surface expression of activating receptors (NKG2D, NKp30, NKp44), producing anti-tumor cytokines in the presence of OC cells and mediating direct tumor cytotoxicity against ascites-derived, primary OC cells obtained from autologous patients. This discovery shows a possibility to create cytotoxic NK cells from the MA of OC patients, which shows a hopeful immunotherapeutic target for the second-line treatment of OC [116].

#### 3.1.2. Adaptative Immune Cells in the Ovarian Cancer Tumor Immune Microenvironment

Tumor Infiltrating Lymphocytes (TILs)

TILs are a type of white blood cells, which include T cells or B cells localized in the tumor and grouped in islets (intraepithelial) and in the peritumoral space (stromal) [117,118]. TILs can be found in primary tumors and omental metastases [119,120,121,122,123,124,125,126,127] and their presence has been correlated with a good prognosis, attributable to its role in the control of tumor growth by activating anti-tumor immune response [121]. Antitumoral responses were mostly characterized by the secretion of TNFα and GM-CSF [125]. Zhang and colleagues showed that intraepithelial CD3^+^ TILs can be found in >50% of advanced-stage EOC with their presence correlating with a five-year overall survival rate of 38% in contrast to 4.5% in patients whose tumors do not have T cells [128]. Even after the debulking and platinum-based chemotherapy, the presence of intraepithelial CD3^+^ TILs increased the five-year overall survival rate (>70%) in comparison to patients whose tumors do not contain TILS (11%) [129]. T cell-rich tumors are correlated with better overall survival (OS) and were associated with the increased expression of IL-2, IFN-γ, and lymphocyte-attracting chemokines within the tumor such as CXCL9 [118], CCL21, and CCL22 [130]. On the contrary, tumors without TILS were associated with an increased level of VEGF, an angiogenic regulatory factor in the TIME associated with early recurrence and short survival [131]. An investigation of the composition of TILS in patients with OC at different stages by Fialova et al. showed that early stages were characterized by a strong Th17 immune response followed by Th1 recruitment for stage II [129]. Among infiltrating T cells, CD8^+^ T cells are associated with a better prognosis, while CD4^+^ T cells expressing the transcription factor FOXP3 (a marker of Tregs) suppress the beneficial effects of CD8^+^ T cells in the TIME [88,111,132,133,134,135,136]. Helios^+^ (a hematopoietic-specific transcription factor involved in the regulation of lymphocyte development) activated Tregs, and high quantities of myeloid dendritic cells and monocytes/macrophages were detected in the advanced stages III and IV [133]. Another study proved that the intratumoral accumulation of CXCR3 ligands, such as CXCL9 and CXCL10, predicts doubled overall survival in advanced HGSC [137]. This study also identified PGE2 as a negative regulator of chemokine secretion that contributes to tumor progression by blocking TILs recruitment in OC [137]. Other researchers showed that reduced EOC patient survival is related to the expression of both COX-1 and COX-2, which are negatively correlated with intraepithelial CD8^+^ TILs [138]. Some studies showed that improved disease-specific survival for EOC patients is correlated with the presence of both intraepithelial CD4^+^ and CD8^+^ T cells [139,140]. However, other studies show that this helpful characteristic is attributed only to intraepithelial CD8^+^ TILS [141]. In 2012, a meta-analysis of 10 studies gathering 1815 OC patients proved the prognostic value of intraepithelial CD8^+^ TILs in EOC specimens regardless of the tumor grade, stage, or histologic subtype [129]. The presence of this specific TILs population in cancer tissues suggests that they spontaneously activated antitumoral responses to control tumor outgrowth [129]. The presence of tumor-reactive antibodies and T cells in the peripheral blood of advanced-stage EOC patients [142,143,144] and oligoclonal tumor-reactive T cells, isolated from blood, MA, and tumors, show the antitumoral response activated by TILs [145,146,147,148]. On the other hand, the lack of intraepithelial TILs is significantly associated with poor survival among EOC patients [129]. At present, immunotherapies aiming to increase the effector functions of pre-existing antitumoral CD8^+^ TILs and triggering effector T cell-trafficking to the TIME are the big goals of cancer immunotherapy [118].

CD8^+^ T Lymphocytes

CD8^+^ T lymphocytes, also known as cytotoxic T lymphocytes, are specialized in killing virus-infected cells and tumor cells. They secrete perforin which creates pores in the plasma membrane of target cells, as well as granzyme, a serine protease that activates caspases and leads to cell apoptosis.

After antigen presentation, naïve CD8^+^ T cells are activated and differentiate into multipotent memory stem cells, which progressively differentiate into memory T-cell (TM) subpopulations and eventually effector CD8^+^ T cells [149]. This sequential differentiation process is supported by a stepwise loss of plasticity, proliferative potential, and capacity for homing into lymphoid organs. In parallel, these cells acquire cytotoxicity, the production of the proinflammatory cytokines IFNγ and TNFα, tropism for inflamed tissues, and eventually a senescent phenotype [149,150]. TM cells are divided into central memory (TCM) and effector memory (TEM) cells, according to different homing and functional properties [151]. TCM cells express lymph node homing receptors (CCR7 and CD62L), produce lower levels of IFNγ in response to antigen presentation, and differentiate into TEM cells upon secondary stimulation. In contrast, TEM cells express receptors for chemotaxis into inflamed tissue (CXCR3) and rapidly secrete higher levels of effector cytokine IFNγ after memory stimulation with an antigen [151]. Besides these specificities, CD8^+^ T cells change in their dependence on glycolysis and oxidative phosphorylation as their energy source [149,152,153]. Activated CD8^+^ T cells rewire their metabolism to achieve the energetic and anabolic requirements to support their rapid proliferation and effector function, so they increase glucose and glutamine uptake, aerobic glycolysis, oxidative phosphorylation, and glutaminolysis, while they suppress fatty acid oxidation. Moreover, the tricarboxylic acid cycle is also used as a source of intermediates for nucleotide, protein, and lipid synthesis [50,154,155]. Several studies report that the presence of intra-tumoral T cells in ovarian carcinoma decreases the rate of recurrences and prolongs patient survival [18,34,131,156], indicating that T cells contribute to a more efficient eradication of tumor cells. In addition, the OC environment damages the anti-tumor function of CD8^+^ T cells through the inhibition of signaling pathways, including checkpoints triggered by cytotoxic T lymphocyte antigen 4 (CTLA-4), programmed cell death protein 1 (PD1), lymphocyte activation gene 3 protein (LAG-3), and T-cell immunoglobulin and mucin-domain containing protein 3 (TIM-3) [157,158,159,160]. These suppressive signals are delivered by ligands expressed on antigen-presenting cells, tumor cells, and tumor-infiltrating immune cells. The induced suppression can be efficiently blocked by antibodies, including anti-CTLA-4, anti-PD1, and anti-PD-L1 [161,162]; nevertheless, the specific contribution of immune checkpoints to OC progression is doubtful. The checkpoint blockade by anti-CTLA4 antibody [157] or anti-PD1 antibody [159] showed limited clinical efficacy and it is already known that the over-expression of PD-L1 in MA tumor cells is associated with CD8^+^ T cells malfunction and reduction in the anti-tumoral response [163]. The expression of PD1 is also a characteristic of T cell exhaustion, detected by low levels of secretion of cytokines by T-cells and its loss of cytotoxicity, which allows its elimination from the TIME [164]. The suppressive effect of MA involves a reduced T cell receptor (TCR) signaling and the activation of the transcription factors NF-κB and NFAT, which are crucial for T cell activation, resulting in the inhibition of signal transduction upstream of phospholipase C-γ (PLCγ) [165]. Nevertheless, the infiltrating CD8^+^ T cells could maintain their functionality because it was proven that the increased levels of IFNγ in tumor tissue of patients with favorable clinical outcomes are maintained. Further, MA play an essential role in T cell biology, thus favorable (i.e., recruitment of T cells into the tumor) and unfavorable (i.e., inhibition of activation-associated signal transduction pathways) aspects [166]. The utilization of these cytotoxic cells through improved immune therapies could be a way to improve the prognosis of OC patients [30,166].

CD4^+^ T cells

T helper (Th) cells are CD4^+^ naïve T lymphocytes activated by antigen-presenting cells (APCs). CD4^+^ and CD8^+^ T cells can specifically recognize tumor-associated antigens from cancer cells. CD4^+^ T cells provide cytokine support for CD8^+^ T cell proliferation and expansion to destroy cancer cells and trigger antitumoral responses [118]. In the context of OC, in this review, we will focus only on these four subtypes of T helper cells: Th1, Th2, Th17, and Tregs, which are defined by the cytokine environment and interactions with APCs. These different subtypes stimulate the immune system, even though they have differences in their cytokine production and target cells. Th1 cells are associated with proinflammatory responses important to killing intracellular parasites and spreading autoimmune response by secreting IL-2, IL-3, IFNγ, TNFα, and GM-CSF [167]. They are also responsible for the activation of CD8^+^ T cells or anti-tumoral macrophages. On the contrary, Th2 cells are associated with anti-inflammatory response, immunoglobulin E, and eosinophilic responses by IL-4, IL-5, and IL-13 secretion [167] and the activation of TAMS and, therefore, are associated with tumor progression [168]. There is a direct link between Th2 cells and OC because these cells produce specific cytokines (e.g., IL-4) present in MA and are correlated with a poor prognosis in OC [167]. Th17 cells secrete the cytokine IL-17 that has both effects, anti-tumoral and pro-tumoral, and shows strong interactions with Tregs [169], and the chemokines CXCL9 and CXCL10 that recruit TILs [170]. Miyahara et al. demonstrated that OC tumors have Th17 cells and OC cells, and tumor-associated APCs secrete cytokines that could be responsible for the increase in Th17 cells [171]. However, Fialova et al., observed a high amount of Th17 cells only in the early stages of OC [129], so they decrease with the advancing stage and appear to be inversely related to Tregs [172,173]. Furthermore, Leveque et al. proved that IL-2 can trigger the conversion of OC-associated CD4^+^ Tregs into Th17 cells [169]. Recent data showed that a low density of IL-17-producing cells is related to the increased ratio of CD4/CD8 T cells within MA and is associated with compromised survival [174]. Although the presence of Th17 cells has been confirmed in OC, the specific effects of these cells in tumor progression wait to be revealed.

Tregs cells are a suppressor T cell population responsible for the immune tolerance to self-antigens and the regulation of the immune system [30]. These cells can inhibit the activity of effector T cells due to their competition for IL-2, thus causing a decrease in the levels of this T cell growth factor by the production of immunosuppressive factors (such as IL-10 or TGF-β) or through cell-to-cell contact [175]. Further, in the TIME, Tregs can promote angiogenesis via VEGF production [172]. Tregs have also been associated with the immune paralysis of DCs, which contribute to immunosuppression [58,62]. Regarding the accumulation of Tregs in solid tumors and MA, there is evidence of the direct migration of naturally occurring Tregs mediated by the expression of chemokines receptors (CCR4, CCR5, and CXCR1 [176]) by the TIME, and it has been demonstrated that the levels of circulating Tregs are increased in OC patients [177,178]. Tregs are characterized by the expression of CD4, CD25, and FOXP3 and can be attracted by CCL22 secreted by OC cells or TAMs through interaction with CCR4 [88,179] and other chemokine receptors, such as CXCR3 or CCR10 [144,175]. The chemokine CCR10 ligand (CCL28) produced by tumor cells under hypoxic conditions contributes to Tregs infiltration [180]. Importantly, higher levels of activated Tregs were found in MA when compared with blood [181], meaning that tumor-infiltrating Tregs in MA produce higher levels of IL-10, express higher levels of the activation marker CD69, and proliferate at a higher extent than circulating Tregs from the same patients [181,182]. The number of Tregs also correlates with the proportion of epithelial cell adhesion molecule (EpCAM)^+^ cancer-derived epithelial cells in MA [181], and they increase with stage [183]. Peng et al. reviewed the different populations of Tregs and their potential clinical applications in OC [181]. It is now clear that OC cells modulate the phenotype of immune cells, corroborating Alvero et al.’s findings showing the existence of two subpopulations of OC cells with different cytokine profiles: cancer stem cells and differentiated cancer cells [182]. The differentiated cancer cells can increase the production of Tregs, leading to a tolerant microenvironment, inhibiting an immune response, being correlated with poor survival in OC patients [88]. Since Tregs are strongly influenced by microenvironmental regulation, a strategy of reprogramming these cells could be an alternative for OC treatment to be explored.

#### 3.1.3. Cytokines Present in Malignant Ascites

Cytokines mediate key interactions between immune and non-immune cells in TIME. MA have a highly immunosuppressive microenvironment [153] and it was demonstrated in the presence of both the pro-tumorigenic and anti-tumorigenic factors of TIME [41,42,184,185]. The anti-tumorigenic cytokines are secreted by Th1 cells, such as IL-2, IL-3, INFγ, TNFα, CCL4, and CXCL10, and the pro-tumorigenic cytokines are secreted by Th2 cells, including IL-4, IL-5, IL-9, IL-10, IL-13, IL-15, CCL2, and VEGF [156,186]. Figure 4 shows some effects of cytokines in the tumor microenvironment. These cytokines cumulatively contribute to creating a pro-inflammatory and immunosuppressive tumor microenvironment [187].

As mentioned, several interleukins are clearly associated with a Th1 or Th2 response, so significantly lower levels of IL-2 in MAs are consistent with a reduced Th1 response [185], because IL-2 is associated with T cell and NK cell growth factor but blocks T cell responses by the maintenance of Treg cells and induction of activation-induced cell death [188].

Xie and colleagues found that decreased levels of IL-7 (associated with carcinogenesis, immunosuppression, and epithelial–mesenchymal transition [189,190]) appear to have similar increasing effects as IL-2 in MAs compared to serum [189]. IL-7 is also important for lymphocyte survival and is associated with the induction of anti-tumor T-cell response and decreased levels of IL-17 [191,192].

IL-15 (activates lymphocytes to produce IFNγ [191]) binds to the IL-2 receptor and stimulates both antigen-independent expansion and the long-term survival of anti-tumor CD8^+^ T cells [193].

Some studies evaluated cytokine concentrations in MA and found increased levels of inhibitor cytokines such as IL-6 and IL-10 but, in some cases, also found elevated levels of stimulatory cytokines such as IL-1β and TNFα compared to the normal control [184]. IL-1β promotes inflammation-induced carcinogenesis but also recruits antineoplastic immune cells that may block metastatic outgrowth [194,195]. TNFα produces and maintains a network of other mediators that promote tumor growth and peritoneal spread by stimulating the release of IL-6 and other chemokines such as CCL2 and CXCL12, macrophage migration-inhibitory factor (MIF), and VEGF. All these factors may act in an autocrine/paracrine manner to promote the colonization of the peritoneum and neovascularization of developing tumor implants [45].

The cytokines IL-6 and IL-10 are frequently analyzed due to their correlation with poor prognosis and response to therapy [42,46]. IL-6 is associated with MA formation in OC patients being involved in the upregulation of VEGF expression which leads to increased vascular permeability [185], promotes tumor growth, mediates cytokine release, and is associated with cachexia (extreme weight loss and muscle wasting) in cancer patients [196,197,198]. IL-10 promotes cytotoxicity but inhibits anti-tumor responses [199,200]. This cytokine is produced at high levels by MDSCs and plays a role in creating a tumor-permissive microenvironment [201,202], so its blockage improves MDSC-mediated immunosuppression and improves survival because this function is not redundant with other immunosuppressive molecules [202]. The blockage promoted by IL-10 increases cytotoxic T cell function in the peritoneal cavity and limits tumor spread [84]. In MAs are found high levels of IL-4, IL-10, TGF-β, and VEGF [182,183]. IL-4, IL-10, and TGF-β can affect phagocyte function, silencing macrophages, and DC activity [188]. Importantly, TGF-β can be also produced by ovarian cancer cells and is a powerful immunosuppressor within the tumor microenvironment, affecting NK and dendritic cell activity, cytokine production, and T-cell function [203]. The increased secretion of TGF-β within the tumor microenvironment recruits Tregs via the expression of FoxP3 [204], which ultimately results in diminished cytotoxic T-lymphocytes [205].

The overexpression of the proangiogenic chemokine IL-8 has been associated with poor outcomes in OC, enhancing tumor progression [206] by promoting tumor implant neovascularization [184] and attracting neutrophils facilitating a suppressive environment [207].

The chemokines are also important to the recruitment of immune cells into the tumor microenvironment, having distinct effects on tumor progression. The chemokines CXCL10 and CCL4 are found elevated in MAs, with CXCL10 being associated with increased anti-tumor response in a mouse central nervous system model [208] and the intratumoral expression of CCL4 correlated with the inhibition of colorectal tumor growth and tumor-specific CD8^+^ T cells response [209].

The neutralization of the chemokine CCL2 results in reduced tumor burden in a mouse prostate cancer model [210]. On the other hand, CCL5 is a chemokine that appears to induce the activation of NK cells and enhances anti-tumor immunity in a mouse model [211]. PDGF regulates cell growth and division with a significant role in angiogenesis, inducing VEGF production in OC cells, which leads to the correlation between PDGF-BB and VEGF expression in MAs [212]. The overexpression of VEGF has an important role in OC progression and blocking this pathway appears to improve survival [213,214].

Evaluating the influence of chemokines in OC, we could conclude that the overexpression of VEGF and CCL2 and the reduction in CCL5 in MA may provide a facilitating pathway for tumor dissemination in the peritoneal cavity [215].

Giuntoli L. Robert and colleagues compared the levels of 27 cytokines and chemokines in MAs and the serum of OC patients and found significant differences between these two samples. They found statistically significant elevated levels of IL-6, IL-8, IL-10, IL-15, CXCL10, CCL2, CCL4, and VEGF and significantly reduced levels of IL-2, IL-5, IL-7, IL-17, platelet-derived growth factor (PDGF)-BB, and CCL5 in MA compared to serum, concluding that MAs are an inflammatory microenvironment [153].

In conclusion, different cytokines and chemokines are associated with the prognosis in OC. IL-2, IL-5, IL-7, and CCL5 are associated with a better prognosis, and IL-6, IL-8, IL-10, CCL2, and VEGF are associated with a worse prognosis. All of them are putative biomarker candidates in the OC context.

## 4. Concluding Remarks and Future Perspectives

The majority of OC patients are diagnosed at an advanced stage (stage III/IV) with metastasis within the pelvic and peritoneal cavities, accumulating large volumes of MA comprising a mixture of tumor and non-tumor cells. In this metastatic niche, MA play an essential role in the OC dissemination process and are present in patients at several stages of the disease progression. This liquid tumor microenvironment is rich in immune cells that interact with tumor cells and this crosstalk has a high impact on tumor progression. Thus, it is crucial to understand the mechanisms that support tumor dissemination in this liquid metastatic microenvironment, with particular attention given to the role of immune cells in this environment.

In this review, we showed that some cytokines and chemokines are associated with good prognosis in OC including IL-2, IL-5, IL-7, and CCL5. On the contrary, IL-6, IL-8, IL-10, CCL2, and VEGF are associated with a worse prognosis in these patients. The immune cells also play an important role in MA, including cells from innate and adaptive immune systems. Many of these cells (TAMs, NK cells, MSDCs, and Tregs) are dysregulated in the MA and are associated with immune suppression, chemoresistance, and worse overall survival of the patients.

Currently, several studies consider that targeting the immune metabolism could be an alternative treatment in OC patients. The current principle of immune metabolism considers that cancer cells are inserted in a context of multiple cell types, including the immune type. In fact, several metabolic drugs used in other disease contexts are being tested to target OC cell metabolism [50]. These compounds showed to have an immunotherapeutic effect when used alone or in combination with other drugs such as immune checkpoint inhibitors [50]. The understanding of these processes and the mechanisms responsible for their perturbation in individual patients could be a way to improve OC management.

## Figures and Tables

**Figure 1 ijms-23-10692-f001:**
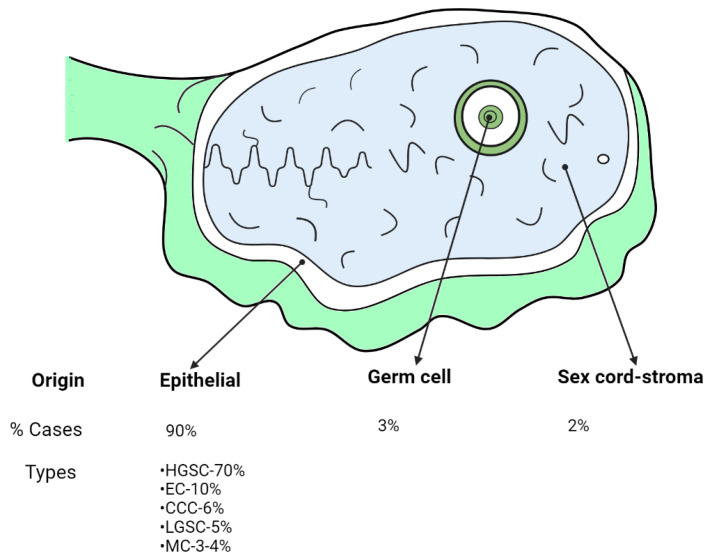
The different origins of ovarian tumors, specifically the types of Epithelial Ovarian Carcinoma (EOC), created by Biorender.com (accessed on 26 August 2022). High-grade serous carcinoma (HGSC); Low-grade serous carcinoma (LGSC); Endometrioid carcinoma (EC); Clear cell carcinoma (CCC); Mucinous carcinoma (MC).

**Figure 2 ijms-23-10692-f002:**
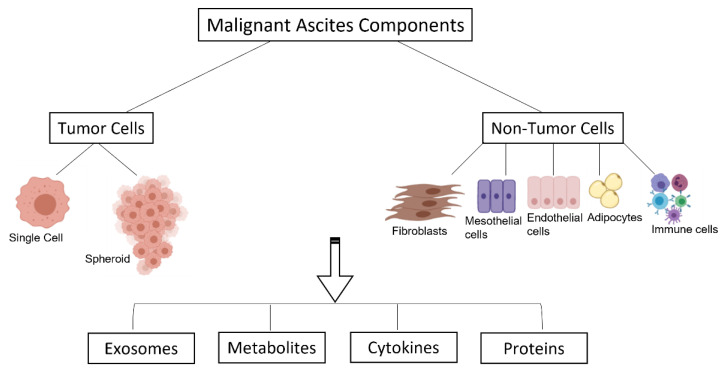
Cellular and acellular components of Malignant Ascites, created by Biorender.com (accessed on 26 August 2022) and adapted from Kim et al. [2]. MAs are composed of tumor cells (single cells or as spheroids) and non-tumor cells, including fibroblasts, mesothelial cells, endothelial cells, adipocytes, and immune cells. These types of cells communicate with each other through acellular factors, including cytokines, proteins, metabolites, and exosomes.

**Figure 3 ijms-23-10692-f003:**
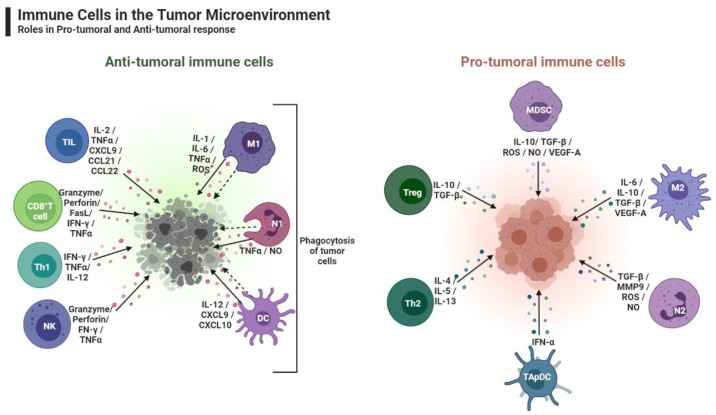
The role of Immune Cells in the tumor microenvironment, created by Biorender.com (accessed on 26 August 2022). On the left is represented the immune cells that act as tumor killers by the production of cytokines or by phagocytosis, that can destroy tumor cells, and on the right, the cells that contribute to immune suppression. TIL- Tumor Infiltrating Lymphocytes; CD8^+^ T—cytotoxic T cells; Th1—CD4^+^ helper 1 T cell; NK—Natural Killer; M1—Macrophage Anti-tumoral; N1—Neutrophil like-type 1; DC—Dendritic cell; Treg—Regulator T cell; Th2—CD4^+^ helper 2 T cell; TApDC—Tumor-Associated plasmacytoid Dendritic cell; N2—Neutrophil like-type 2; M2—Macrophage Pro-tumoral; MDSc—Myeloid-Derived Suppressor Cells.

**Figure 4 ijms-23-10692-f004:**
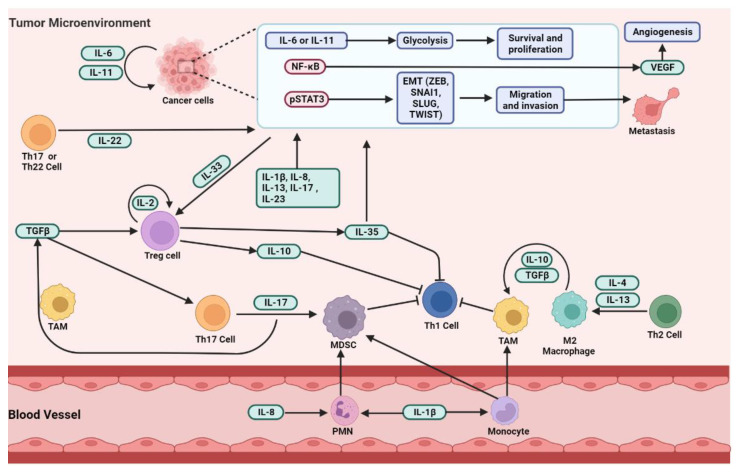
The effect of cytokines in the tumor microenvironment, created by Biorender.com (accessed on 26 August 2022), adapted by Briukhovetska et al. [185]. Immune evasion and tumor progression depend on cancer cell-intrinsic and -extrinsic cytokine signaling. It is shown that OC overexpresses certain cytokines, for example, IL-6, which act in an autocrine way to upregulate glycolysis and induce metabolic reprogramming, nuclear factor kappa-κB (NF-κB), and signal transducer and activator of transcription 3 (STAT3). These pathways in change can lead to epithelial–mesenchymal transition (EMT), increased proliferation, reduced apoptosis, increased migration, and the production of cytokines, such as IL-8 and VEGF, which induces angiogenesis. However, other cytokines, such as IL-1β, IL-13, IL-17, IL-22, IL-23, and IL-35 also induce EMT and, thus, tumor progression. Tumor-secreted IL-8 stimulates the recruitment of polymorphonuclear leukocytes (PMNs) and, in association with monocytes, they differentiate into MDSCs, inhibiting Th1 responses. MDSCs, TAMs, and M2 macrophages that are polarized by Th2-type cytokines contribute to higher levels of TGF-β that form an immunosuppressive microenvironment. In sequence, TGF-β together with IL-33 promotes the differentiation of Treg cells, which have a high affinity to IL-2 receptor (IL-2R) and are a major source of IL-10 that, under chronic conditions, suppresses antitumor responses. Additionally, TGF-β with IL-6 promotes the differentiation of Th17 cells that produce IL-17 and promote more MDSC recruitment and differentiation. pSTAT3, phosphorylated STAT3; EMT transcription factors (ZEB, SNAI1, SLUG, TWIST).

## Data Availability

The authors confirm that the materials included in this chapter do not violate copyright laws. Where relevant, appropriate permissions have been obtained from the original copyright holder(s), and all original sources have been appropriately acknowledged or referenced.

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
