# Peer review of "Immune Tumor Microenvironment in Ovarian Cancer Ascites"

_ijms, 2022, doi:10.3390/ijms231810692_

Round 1
Reviewer 1 Report
My dears,
Please find attached your manuscript with my comments.
Overall the review is clear and I enjoyed reading it. My major comment would be to try and not deviate from the topic. Your focus is represented by malignant ascites so the intratumoral environment should be left aside. I would also focus on ascitic fluid markers that predict therapy outcomes since the stratification of OC risk groups is to be improved. Also about intraperitoneal therapies (HIPEC, PIPAC...), any studies regarding immune markers in such therapies?
I hope you will find my feedback useful
Best wishes!

Author Response
We appreciate the time and effort that the reviewers dedicated to providing feedback on our manuscript. We have incorporated the suggestions made by the reviewers in our manuscript and addressed some of the reviewer’s concerns below.
In this review paper, our main objective was to discuss the immunological biomarkers present in the ascitic fluid that are capable of predicting ovarian cancer patients’ outcomes. In fact, we expect to find biomarkers that could improve the actual patient stratification.
Regarding intraperitoneal therapies and studies addressing immune markers in such therapies, the records are still scarce but we believe that this theme will be extensively discussed in the following years.
We hope that we answer all your questions. Thank you very much for the positive feedback!
Best Regards,
Sara Ricardo
Reviewer 2 Report
This review provides a comprehensive overview about immune tumor microenvironment in ovarian cancer ascites. Please see below comments to improve the clarity of the manuscript.
1. Lane 111. Please address whether ascites is more common in HGSC than other type and the mechanisms about how to produce ascites.
2. Figure 4: The TGFβ could be produced by ovarian cancer cells. Please cite more specific review articles about cytokines cross-talk between cancer cells and immune cells in tumor microenvironment.
Author Response
We appreciate the time and effort that the reviewers dedicated to providing feedback on our manuscript. We have incorporated the suggestions made by the reviewer in our manuscript which is transcribed below.
Ascites is a condition that is more common in HGSC than in other ovarian cancer types. We included the following paragraph in the potin 3. of the Ms:
”Ascites can occur in different diseases, such as cirrhosis, pancreatitis, nephritis, heart failure, and cancer. Malignant ascites is pathological accumulation of fluid in the peritoneal cavity that can be found in many neoplasms of peritoneal organs (ovarian, endometrial, pancreatic, gastric, colorectal and liver). This inflammatory condition is a consequence of a disruption in the balance of fluid production and reabsorption being triggered by an increased vascular permeability, peritoneal lymphatic obstruction, and high levels of fluid production. The presence of ascites is often indicative of tumor cells in the peritoneal cavity or peritoneal carcinomatosis.”
Cytokine TGF-β can be, in fact, produced by ovarian cancer cells and several studies have been focused on understanding its oncogenic activity. Also, it is well known that this cytokine plays an important role in ovarian cancer and is a powerful immunosuppressor within the tumor microenvironment, affecting NK and dendritic cell activity, cytokine production, and T-cell function. One important mechanism to suppress cytotoxic T-cell function is through the increased production of Tregs. Increased secretion of TGF-β within the tumor microenvironment recruits Tregs via expression of FoxP3, which ultimately results in diminished cytotoxic T-lymphocytes. We included text and more specific review articles about this specific cytokine in the section 3.1.3: “Importantly, TGF-β can be also produced by ovarian cancer cells and is a powerful immunosuppressor within the tumor microenvironment, affecting NK and dendritic cell activity, cytokine production, and T-cell function. Increased secretion of TGF-β within the tumor microenvironment recruits Tregs via expression of FoxP3, which ultimately results in diminished cytotoxic T-lymphocytes.” New references were also included and highlighted in the revised version of the Ms.
We hope that we answer all your questions. Thank you very much for the positive feedback!
Best Regards,
Sara Ricardo
Reviewer 3 Report
The review article entitled ' Immune tumor microenvironment in ovarian cancer ascites’ by Diana Luísa Almeida-Nunes et al, greatly describes the various immune cells that orchestrate the tumor microenvironment in ovarian cancer. The article is very interesting and provides an overview. However, The authors could improve the article.
1. The authors need to describe the current advances in the field. Any clinical trials that were performed in this context.
2. Any potential targets that could be targeted for the treatment of the disease.
3. With the emergence of CAR-T therapy and neoantigens and other therapeutic advances the authors need to shed light on this topic in a separate section.
4. Few typographical errors are observed like on page 3, line 123. The spelling should be hepatic bile.
5. The authors need to follow the numbering pattern throughout, On page 5, the sub section of 3.1.1 is not numbered.
6. Abbreviation of TIME on line 183 needs to be described.
Author Response
We appreciate the time and effort that the reviewers dedicated to providing feedback on our manuscript. We have incorporated the suggestions made by the reviewer in our manuscript which is transcribed below.
In this review paper, our main objective was to discuss the immunological biomarkers present in the ascitic fluid that are capable of predicting ovarian cancer patients’ outcomes. In this manuscript, our focus was not to include therapies such as target therapies, immunotherapies (including CAR-T therapy), and clinical trials because this is an extensive theme that we plan to address in a discussion of our research results in a future original paper.
The errors identified in the manuscript were corrected and are highlighted in the reviewed Ms version.
We hope that we answer all your questions. Thank you very much for the positive feedback!
Best Regards,
Sara Ricardo